## COMMENT

# Melvin is a conversational voice interface for cancer genomics data

Akila R. Perera [1,2], Vinay Warrier[1], Shwetha Sundararaman[1], Yi Hsiao [1,3],
Soumita Ghosh[1], Linganesan Kularatnarajah[1] & Jason J. Pitt [1,4,5✉]

Despite large collections of cancer genomics data being openly available, the inability to quickly interrogate this information remains a barrier for researchers and oncologists. Here we present Melvin, an Amazon Alexa skill to explore cancer genomics data through simple conversations.

Cancer genome sequencing initiatives have generated petabytes of data across tens of thousands of samples. While this has spurred multiple challenges in data processing and warehousing, the majority of those who consume cancer genomics data – namely researchers and clinicians – need efficient ways to perform basic queries and analyses. There is a high demand for intuitive tools to explore downstream results stemming from projects such as The Cancer Genome Atlas (TCGA)[1]. Web-based graphical user interfaces (GUIs) have been developed to address this need[1,2]. However, for many users, standard GUIs lack accessibility and can require minutes to answer questions such as, "What percentage of TCGA breast cancer patients have *TP53* mutations?" Modern interfaces – particularly those leveraging augmented intelligence – show promise to streamline inquiries, democratize analytics, and enhance digital health applications in cancer genomics[3,4].

Voice user interfaces (VUIs) are revolutionizing how we access information and perform tasks. VUIs have multiple advantages over GUIs. 1) A minimal learning curve makes them convenient for a broad user base. 2) Their conversational nature allows queries to be resolved both quickly and progressively. For both English and Mandarin, users are able to provide input nearly three times faster through speech-to-text than manual typing[5]. 3) They are readily accessible via mobile phones, computers, and smart home devices. Within the biosciences, VUIs have been developed only for basic information retrieval (e.g. gene definitions)[6] or managing laboratory operations[7,8]. These tools do not retain the context necessary to progressively answer deeper scientific questions. However, when designed to be conversational, VUIs provide an opportunity for anyone to query complex data – such as cancer genomics – in real time using natural language.

## Results

Here we present Melvin, a VUI to explore and analyze cancer genomics data using any Amazon Alexa-capable device (e.g. mobile phones, tablets, Amazon Echo, etc.). Most supported queries involve a combination of three key attributes: GENE, CANCER TYPE, and DATA TYPE. In Fig. 1a, we demonstrate how individuals can leverage multi-turn conversations in Melvin to obtain the *TP53* mutation rate in TCGA breast cancer. A user begins by saying "Tell me about mutations," followed by, "Show me breast cancer." After processing these requests through four interoperating cloud-based microservices (Fig. 1b; Supplementary Fig. 1a, b and Supplementary Table 1 and "Methods" section), Melvin responds by indicating the most

[1]Cancer Science Institute of Singapore, National University of Singapore, Singapore, Singapore. [2]School of Computing, National University of Singapore, Singapore, Singapore. [3]Department of Computational Medicine and Bioinformatics, University of Michigan, Ann Arbor, MI, USA. [4]NUS Centre for Cancer Research, Yong Loo Lin School of Medicine, National University of Singapore, Singapore, Singapore. [5]Genome Institute of Singapore, Agency for Science, Technology and Research (A*STAR), Singapore, Singapore. ✉email: jason.j.pitt@nus.edu.sg

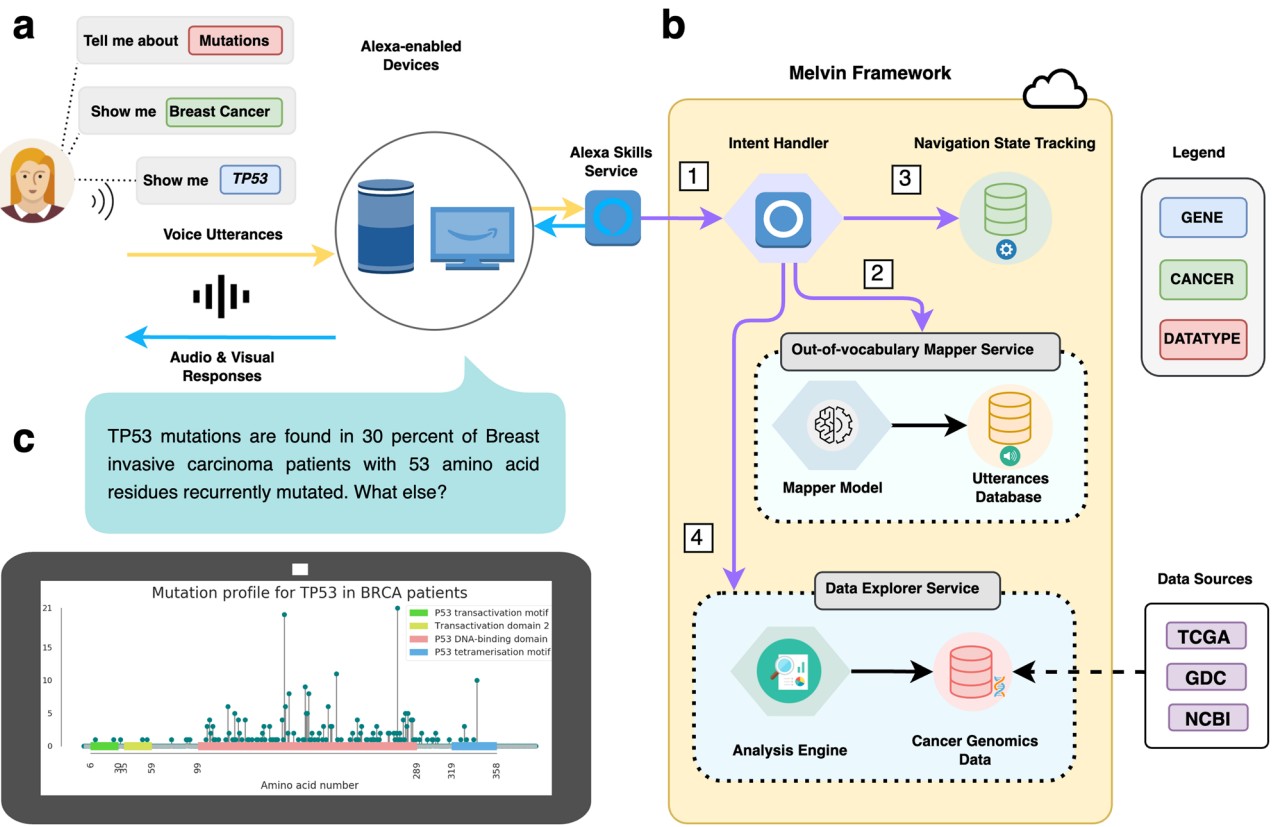

**Fig. 1 Melvin's architecture supports multi-modal responses to common cancer genomics queries. a** Engaging Melvin through an Alexa-enabled device, a user has already expressed two attributes of interest – a CANCER TYPE (breast cancer) and a DATA TYPE (mutations). Next, the user expresses a GENE (TP53). The user's voice utterance is captured by the device and sent to Alexa Skills Service. **b** The transcribed query is received by the Melvin Intent Handler (1) which calls the Out-of-vocabulary Mapper Service (OOVMS) (2) to map the utterance to a supported attribute. The navigation state is updated (3) based on the identified attribute type (GENE) and its value (TP53). The updated navigation state (breast cancer, mutations, and TP53) dictates the real-time analysis request, which is sent to the Data Explorer Service (4). **c** Computed results are sent to the Intent Handler where speech and graphical responses are generated and relayed back to the user. GDC Genomic Data Commons, TCGA The Cancer Genome Atlas, NCBI National Center for Biotechnology Information.

frequently mutated genes in TCGA breast cancer both audibly and visually (Supplementary Movie 1). When this query is further narrowed to *TP53*, Melvin vocalizes the *TP53* mutation rate and displays a detailed graphical summary (Fig. 1c). This entire process takes less than 1 minute and requires no physical interaction – exemplifying Melvin's ability to quickly resolve queries while requiring minimal bioinformatics and UI knowledge from users.

**Cancer genomics knowledge base**. Melvin's knowledge base contains harmonized genomic datasets representing all 33 cancer types from TCGA[9,10]. Users can inquire about mutations (SNVs and/or indels), copy number alterations (amplifications and/or deletions), and gene expression (Supplementary Table 2). As a proof-of-principle, we have integrated mutational and copy number data from the Breast Cancer Somatic Genetics Study (BASIS)[11] to demonstrate Melvin's native ability to support datasets beyond TCGA (Supplementary Movie 2). Ancillary information – such as gene definitions and therapeutic actionability – is available to help contextualize and interpret results (Supplementary Table 3; "Methods" section; and Supplementary Movie 3). Importantly, all results, including high-resolution images, can be instantly emailed to users.

**Conversations and analytics via a finite state machine**. Cancer genomics queries are often difficult to verbalize in a cohesive

sentence. To circumvent this friction, Melvin allows users to provide attributes incrementally and in any order (Fig. 2a). This process is implemented as a finite state machine where Melvin progressively returns biologically relevant responses based on the current state (Supplementary Movie 4). In addition to simplifying interactions via multi-turn conversations, Melvin's state-based design retains context to avoid repetitive, single-turn queries – which are standard for GUI alternatives. For example, after navigating to mutations, breast cancer, and *TP53*, a user may want to replace *TP53* with *PIK3CA*. Instead of building an entirely new query, users can simply say, "How about *PIK3CA*?" Melvin will then return the *PIK3CA* mutation rate for breast cancer and associated visual output in seconds. This rapid switching of values can be done for any attribute type and makes billions of unique queries possible within a single, seamless conversation.

This state-based framework also enables Melvin to support more complex analytical queries. Fig. 2b depicts a user invoking Melvin's *compare* functionality to intersect *PIK3CA* mutations and copy number alterations in breast cancer. *Compare* contrasts two attribute values of the same type (e.g. mutations and copy number alterations) in the context of other attribute types (e.g. *PIK3CA* and breast cancer). Pragmatically, this framework allows users to assess alteration co-occurrence, compare mutational frequencies, and juxtapose putative driver genes between cancer types (Supplementary Table 4). Additionally, we have developed *split-by* which allows users to determine

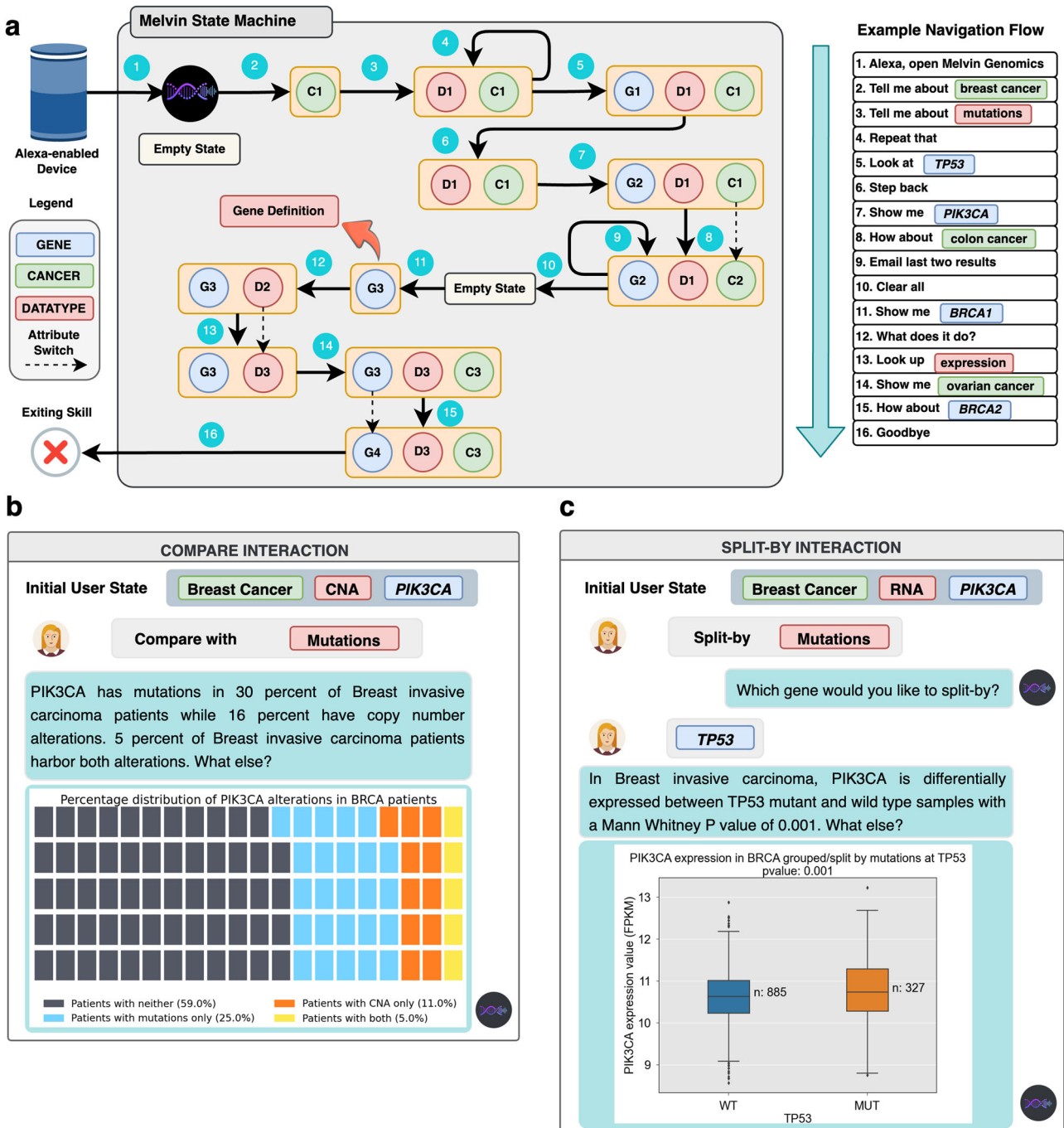

**Fig. 2 Melvin's state-based navigation flow enables genome analytics through multi-turn conversations. a** The Melvin navigation state is composed of three main attributes: GENE, CANCER TYPE, and DATA TYPE. At the beginning of a user session, all three attributes have empty values. Incoming attribute values are used to update the current state and dictate Melvin's response. **b** Example of a *compare* operation. With the context of *PIK3CA* (GENE) and breast cancer (CANCER TYPE), the user expresses the desire to compare two data types - copy number alterations and mutations. **c** Example of a *split-by* operation. The user has navigated to a state containing *PIK3CA* (GENE), breast cancer (CANCER TYPE), and gene expression (DATA TYPE). The user then requests *PIK3CA* expression partitioned mutations in *TP53*.

how a quantitative variable – such as the expression of *PIK3CA* – varies based on a binary variable, like mutations in *TP53* (Fig. 2c; Supplementary Movie 5).

**Crowdsourcing to enhance voice recognition.** Correctly understanding user requests – or utterances – is essential for VUI functionality. Pronunciations for domain-specific terms, particularly gene names, are often out-of-vocabulary (OOV) – making accurate speech-to-text conversion difficult for Amazon Alexa. To

address this, we developed an approach to crowdsource utterances for cancer-associated genes[12], cancer types, and data types (Supplementary Figure 2; Supplementary Data 1, 2; Methods). A dataset of > 24,000 utterances was collected from domain experts of mixed nationalities (see Methods) and subsequently utilized in Melvin's OOV Mapper Service (OOVMS). Incoming user utterances that cannot be mapped to a supported attribute by exact or crowdsourced utterance matching are subject to a random forest classifier (Supplementary Fig. 3a–c). This latter model –

constructed using character, length, and phonetic features of the crowdsourced utterances – predicts the best matching GENE, CANCER TYPE, or DATA TYPE. Overall, Melvin's OOVMS correctly mapped 88.9% of test utterances – a 69.8% improvement over Amazon Alexa's baseline performance (Methods).

To mitigate scenarios where the OOVMS fails to properly map a user's utterance, we developed a web portal where users can submit personal pronunciations (Supplementary Figure 4). Users can specify their intended target (e.g. *KRAS*) and record audio of their own pronunciation (e.g. "keɪ ras"). This tells Melvin, "When you hear me say 'keɪ ras' I mean '*KRAS*.'" Personal pronunciations are tethered to users' Amazon Alexa accounts and can be managed within a simple web-based console.

## Discussion

While the application of voice technology is still in its infancy, Melvin demonstrates its utility within cancer genomics and data analytics. There are multiple opportunities to extend this work. Despite the strong performance of the OOVMS, large-scale crowdsourcing of pronunciations could further improve our machine-learning model and increase the list of attributes Melvin supports. This includes a broader sampling of national and regional accents (see Methods) for gene name pronunciations. Notably, crowdsourced data could be repurposed to enable voice assistants to say gene names using common vocalizations.

In addition to offering faster data exploration through spoken natural language, VUIs may have specific benefits for scientific discussions. They could serve as augmented intelligence agents to help oncologists and bioinformaticians interpret panel sequencing reports within molecular tumor boards (Supplementary Fig. 5)[13]. For example, attendees could engage Melvin to determine if a gene of interest is putatively actionable or frequently mutated in a given cancer type. As voice technology improves, VUIs may be able to capture the context of entire group conversations and play a more proactive role – such as relaying relevant information without an explicit prompt. With its VUI and ability to query databases, Melvin complements other conversational agents such as ChatGPT. In the future, hybrid platforms integrating voice, vetted biomedical databases, and large language models could further enhance digital health applications.

VUIs come with additional security considerations as users' voice recordings could contain private information or be used nefariously (e.g. voice cloning)[14]. Importantly, Melvin only receives users' speech-to-text translations from Amazon Alexa and cannot access user-provided audio data (Supplementary Note 1). However, it is possible that users' voice recordings will be retained by Amazon Alexa itself[15]. We provide details on how this can be circumvented in Supplementary Note 2. Like with any new technology, it is important for users to weigh potential risks prior to use.

Through its state machine approach, Melvin automatically returns analyses and visualizations based on the current attributes and their values (i.e. state). Here, we had to balance analytical detail with concise, conversational responses and fast navigation. A potential limitation of Melvin is its gene-centric design, which may require minor modifications if other genomic elements are to be queryable. Nonetheless, the Melvin framework is extensible and can support more advanced analytics by expanding the number of possible attributes and intents. Key aspects of Melvin's codebase have been open-sourced to encourage communal development (see Software Availability). Additionally, we will continue to augment the operations Melvin can perform in response to user feedback.

As demonstrated with the addition of BASIS, Melvin has been designed to ingest other cancer genomics datasets. We will continue to add high-value datasets (e.g. ICGA PCAWG[16]) over time. With a large Internet of Things footprint, Melvin could provide a complementary interface to cancer genomics datasets hosted by GDC[9], cBioPortal[2], or Xena[1]. As these data warehouses continue to grow, innovative solutions are required to provide on-demand insights. By enabling users to formulate queries through natural language, Melvin helps promote data democratization, scientific discovery, and clinical translation in cancer genomics.

## Methods

**Cloud architecture and design.** In addition to the custom Interaction Model (Supplementary Notes 2 and 3), Melvin consists of four, loosely-coupled microservices deployed on Amazon Web Services (AWS): Intent Handler; Navigation State Tracking, Out-of-Vocabulary Mapper Service (OOVMS); and Data Explorer Service (Supplementary Fig. 1a, b and Supplementary Notes 4 and 7). A serverless computing model is used to build and host the Data Explorer Service and OOVMS components under the AWS Lambda platform. This allows Melvin to be instantaneously scalable without manual intervention. The serverless-framework toolkit packages multiple system components (e.g. OOVMS and Data Explorer Service) into bundles that can be deployed via AWS Cloud Formation platform. Data contained within the Data Explorer Service are structured within Amazon Aurora relational databases. Amazon DynamoDB is utilized to manage state-based tracking, crowdsourced utterances, and personal pronunciations. For additional details on result caching, message passing, and application programming interfaces (APIs), please see Supplementary Notes 4 and 7.

**Cancer genomic data sources.** Melvin's Data Explore Service contains a collection of genomic datasets from the 33 cancer types within The Cancer Genome Atlas (TCGA). DNA-based alteration data – mutations and copy number alterations – were downloaded from the genomics Data Commons (GDC). Mutational calls were taken from four different variant callers: MuTect2, Somatic Sniper, Varscan, and Muse (Data release: 8.0). For a mutation to be included in Melvin, it needed to be called by two or more callers and assigned a MODERATE or HIGH functional impact according to Variant Effect Predictor (VEP). As indels were only called by MuTect2 and Varscan, we required each to be called by both tools to be classified as a mutation in the Data Explorer Service. Per-sample, genic CNAs were also downloaded from the GDC Portal. RNA expression data is from UCSC Toil RNAseq Recompute Compendium (version: 2016-04-12). BASIS SNVs and indels (mutations) were downloaded from the ICGC portal. The `impact_id` for each mutation was mapped to an Ensembl Variant Effect Predictor (VEP) `consequence_type`. Only MODERATE and HIGH impact mutations from each gene's longest transcript (Ensemble v75 GTF) were retained. The union of SNVs called by CaVEMan and the PCAWG Consensus SNV-MNV caller – as well as the union of indels called by Pindel and the PCAWG Consensus INDEL caller – were made queryable by Melvin. Copy number alteration data (`copy_number_somatic_mutation.BRCA-EU.tsv.gz`) was downloaded from the ICGC portal. For each sample, genes (`gene_affected`) were annotated as loss/deletion, neutral, or gain/amplification using the `mutation_type` field. Only BASIS samples (n = 344) that had both mutation and copy number alteration data were made available via Melvin. All genomic datasets were filtered to only include genes from the consensus coding sequence (CCDS) project (version: 22). *Homo sapien* gene definitions and locations are sourced from the National Centre for Biotechnology Information (NCBI) (File: `Homo_sapiens.ags.gz`; accessed May 2019). The therapeutic

actionability of genes within oncology – as well as the associated pharmacologic agents – was taken from the US Food and Drug Administration's (USFDA) Table of Pharmacogenomic Biomarkers in Drug Labeling (accessed July 2021). Peptide and protein domain information for supported *Homo sapien* genes were taken from Pfam (accessed 1 Oct 2019).

**Crowdsourced and custom utterances**. Using our Pronunciation Quiz skill (Supplementary Figure 2; Supplementary Note 6), we crowdsourced utterances (i.e. pronunciations) for genes within the Cancer Gene Census (Supplementary Data 1), cancer types within TCGA and potential synonyms, as well as genomic data types and their potential synonyms ($n = 897$) (Supplementary Data 2) from 9 domain experts at the Cancer Science Institute of Singapore. This group was composed of individuals from different ethnicities (Singaporean, Indonesian, Sri Lankan, American, and Indian) – adding accent diversity to the collected data. 24,093 utterances were collected in total. On average, each expert provided 3 pronunciations for each of the aforementioned domain-specific terms. Due to the time required to generate such training data, we limited our gene pronunciations to the most cancer-relevant genes found within the Cancer Gene Census. However, we also developed the Custom Pronunciation Web Portal (Supplementary Note 6) to allow users to utilize Melvin to explore genes of interest that are not within the Cancer Gene Census. Taking the user pronunciations of the domain-specific terms, a crowdsourced database was populated with attribute values, ASR transcriptions, and attribute types (eg. {*MYC*, "MICK", GENE}).

**Out-of-vocabulary mapper service design and testing**. The OOVMS is a component in the Melvin system that helps to accurately identify the domain-specific term or phrase uttered by the user. The voice request from the user – query utterance – is captured by Alexa and its speech transcription is provided for further processing. These query utterances undergo a chain of three, sequential pipelines: (1) exact matching; (2) crowdsourced utterance mapping; and (3) predicted matching (Supplementary Fig. 3a–c).

In the exact matching pipeline, the incoming query utterance is cleaned, case transformed, and lemmatized to match exact entities (GENE, CANCER TYPE, or DATA TYPE) from a lookup. In cases where the least disambiguation is required, this stage should yield results, thereby foregoing subsequent pipelines. The next pipeline, crowdsourced utterance matching, exploits the database of crowdsourced utterances generated via Pronunciation Quiz (Supplementary Fig. 2 and Supplementary Note 6). The performance of this pipeline is dependent on the distribution of attribute pronunciation variations captured in the crowdsourced utterance database. Any query utterance that cannot be resolved by exact matching is subjected to a lookup within the crowdsourced utterances. If the query utterance can be mapped to a single attribute value via crowdsourced utterances, that attribute value is returned by the OOVMS.

Finally, any query utterance that cannot be resolved by the first two pipelines is subjected to predicted matching. A machine learning (ML) model trained on 90% ($n = 21,683$) of the crowdsourced utterance corpus is prepared to predict an optimal match for the incoming utterance. The preparation of the ML model consists of three steps: (i) dataset preparation, (ii) feature engineering, and (iii) model building. In dataset preparation, basic preprocessing is carried out on the utterance corpus. This involves cleaning and case transformation. In feature engineering, the raw dataset is transformed into flat features that can be used for training an ML model. The utterances are transformed into a combination of representations that are generated using traditional NLP (n-grams and TF-IDF) and context-aware vector representations (word embeddings). Phonetic representation and syllables are also captured in the final set of features generated. In the final step of model building, a random forest classifier (Scikit Learn) is trained – using the aforementioned features – to learn the mappings between the crowdsourced utterance transcriptions and intended target terms. A random forest classifier is a set of decision trees, each built over a random extraction of features and dataset observations. When subjected to a query utterance, it then combines the predictions from all trees, and the final result is calculated by a majority vote (with ties split randomly) (Supplementary Figure 3c).

The remaining 10% ($n = 2,410$) of the crowdsourced utterance corpus was subjected to the dataset preparation steps mentioned above and used for testing. We then determined the percentage of test utterances that could be accurately mapped by Amazon Alexa's native ASR transcriptions versus our three-step OOVMS.

**End-to-end testing framework**. In addition to high coverage to unit testing, we developed a comprehensive end-to-end testing framework to continuously ensure microservice function, proper utterance mapping, and the veracity of Data Explorer Service results. See Supplementary Notes 8 and 9 for further details.

**Statistics and reproducibility**. Statistical testing in Melvin is performed using the numpy (version: 1.18.1) and scipy (version: 1.5.2) Python libraries. Melvin's visualizations are generated using the Python libraries seaborn (version: 0.11.2), matplotlib (version: 3.1.3), squarify (version: 0.4.3), and pywaffle (version: 0.6.1).

**Reporting summary**. Further information on research design is available in the Nature Portfolio Reporting Summary linked to this article.

## Data availability

All data within Melvin's Explorer Service was taken from publicly available sources. Details of these sources as well as dataset release versions can be found in the Methods section.

## Code availability

Melvin is freely available as an Amazon Alexa skill (https://www.amazon.com/dp/B09NZSRBNS). Codebases underlying the Melvin skill (https://github.com/pittlab-genomics/Melvin_Alexa_Skill; https://doi.org/10.5281/zenodo.10118441)[17], Melvin intent handler (https://github.com/pittlab-genomics/Melvin_Alexa_Intent_Handler; https://doi.org/10.5281/zenodo.10118447)[18], Pronunciation Quiz skill (https://github.com/pittlab-genomics/Pronunciation_Quiz_Alexa_Skill; https://doi.org/10.5281/zenodo.10118435)[19], and Pronunciation Quiz intent handler (https://github.com/pittlab-genomics/Pronunciation_Quiz_Intent_Handler; https://doi.org/10.5281/zenodo.10118425)[20] are publicly available as Git repositories as well as a Zenodo repositories. Additional Melvin use cases and animated video demonstrations can be found here: https://www.melvin.pittlabgenomics.com/.

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

## Acknowledgements

J.J.P. is supported by the National Research Foundation Singapore and the Singapore Ministry of Education under its Research Centers of Excellence initiative. YH was supported by the HLA-TW Internship Scholarship. The computational work for this article was partially supported by the National Supercomputing Center, Singapore (https://www.nscc.sg). The authors would like to thank Amazon Web Services and AlexaEDU – particularly Russ LeWinter and Raj Chary – for their technical guidance throughout this project as well as Wu Jialun Andy for genomic data wrangling.

## Author contributions

A.P. designed, implemented, and optimized Melvin's serverless architecture and interaction model. A.P. and V.W. developed the Pronunciation Quiz Alexa skill and Data Explorer Service. V.W. developed the OOVMS. S.S., A.P., and L.K. created the Custom Pronunciation Service. SS established unit and end-to-end testing frameworks. Y.H. developed the early Melvin prototype. Y.H., S.G., and V.W. aggregated and formatted genomic datasets. V.W., S.G., and J.J.P. designed and implemented Melvin's visual output. J.J.P. conceptualized Melvin as well as its UX design, navigation flow, OOVMS, supported intents, and responses. J.J.P. supervised all work and wrote the manuscript with assistance from all authors.

## Competing interests

J.J.P. previously received support from the Amazon Web Services Cloud Credits for Research program. All other authors declare no competing interests.
