## [Peer Review File · Communications Biology]

Reviewers' comments:

Reviewer #1 (Remarks to the Author):

The authors provide an implementation of a conversational voice interface for cancer genomics data, implemented using an Amazon Alexa skill dubbed "Melvin." The skill is focused on access to the TCGA database, but pulls in additional resources to help with context. The authors describe the core utility of the system, and also some specific improvements that were made in adaptation of the interaction models to deal with the challenges of the scientific vocabulary. Additionally, the authors provide a quite comprehensive supplementary information package, including basic details about Amazon Alexa skills, and additional details about the improvement made for handling intents and more customized pronunciation and vocabulary.

Overall, this represents a significant advance in the field. This reviewer appreciates not only the sophisticated framework that the authors have established for querying complex data sets, but also the attention to detail that has been invested in the technical development. In particular, Fig 2, showing the state machine, provides a nice way for the reader to visualize what is happening with Melvin behind the scenes.

While the work has significant value, there are some minor issues with the manuscript that should ideally be addressed before it is published:

1) The url for video demonstrations (www.melvin.pittlabgenomics.com) did not appear to be active. I assume this will be corrected upon publication.

2) A git repository was made available by the authors. This should be highlighted in the manuscript, to show that Melvin is not only freely available to use, but can also be potentially extended by other researchers.

3) More space could be devoted in the main text to the specific use cases for a voice assistant? The idea expressed in Sup Fig 5 of use in a tumor board meeting is interesting that could be expanded on. In other words – when, specifically, do the authors see an advantage to a voice assistant rather than a GUI-based approach? Is it that it can be faster for navigation, or are there other benefits?

4) A bit of an elephant in the room are all the recent advances with Large Language Models are conversational agents (ChatGPT, Bard, etc). Although this work represents a significant contribution without uses of these agents, it may be helpful to consider their impact on future development in the voice assistant space.

5) I find the idea of the pronunciation quiz to be very interesting. Perhaps a further avenue of research to be in repurposing this data (for example, so that other voice assistants correctly pronounce some of these terms in the most logical way).

6) It may be worth noting the limitations of the crowdsourced utterance data, which perhaps overlooks some major categories of accent due to the particular focus (American regional accents, British accents, etc)

7) Does the use of this design of state machine with particular slots make the skill challenging to extend? How would the authors propose extending it to include additional data sources? Please add a few comments about how the skill can be improved in the future.

Thank you for the opportunity to review this research.

Reviewer #2 (Remarks to the Author):

This is an interesting paper that presents Melvin, a voice-based user interface for exploring cancer genomics data. The authors have developed a flexible architecture and conversation model that allows users to query mutation, copy number, and expression data from TCGA through natural language interactions.

Some limitations to consider:

1. The data source is currently restricted to TCGA, so expanding to additional cancer genomics datasets could improve utility. Providing access to raw sequencing data may also be valuable for power users.
2. The accessibility of voice interfaces depends on ownership of Alexa devices. Integrating with web or mobile could increase reach.
3. More advanced statistical and visualizations capabilities could further augment what users can achieve through conversational interactions.
4. Discussion of privacy considerations is limited. As voice data presents unique risks, this should be expanded.

Overall, this is a promising proof-of-concept for conversational AI in genomics. The multi-turn model moves the field forward and the technical innovations around speech recognition are impactful. Addressing some of the limitations around data sources, platforms, analytics, and privacy could strengthen the work further.

Referee expertise:

Referee #1: Computational biology, Alex tool development

Referee #2: Computational genomics

Reviewers' comments:

Reviewer #1 (Remarks to the Author):

The authors provide an implementation of a conversational voice interface for cancer genomics data, implemented using an Amazon Alexa skill dubbed "Melvin." The skill is focused on access to the TCGA database, but pulls in additional resources to help with context. The authors describe the core utility of the system, and also some specific improvements that were made in adaptation of the interaction models to deal with the challenges of the scientific vocabulary. Additionally, the authors provide a quite comprehensive supplementary information package, including basic details about Amazon Alexa skills, and addition details about the improvement made for handling intents and more customized pronunciation and vocabulary.

Overall, this represents a significant advance in the field. This reviewer appreciates the not only the sophisticated framework that the authors have established for querying complex data sets, but also the attention to detail that has been invested in the technical development. In particular, Fig 2, showing the state machine, provides a nice way for the reader to visualize what is happening with Melvin behind the scenes.

While the work has significant value, there are some minor issues with the manuscript that should ideally be addressed before it is published:

We greatly appreciate this Reviewer's kind comments about our work. We have spent a significant amount of time considering and implementing improvements to user interaction/experience as well as debating how technical aspects can be best relayed to a lay audience. It is genuinely encouraging to hear such a positive response from another domain expert.

1) The url for video demonstrations did not appear to be active. I assume this will be corrected upon publication.

We thank the Reviewer for this detailed observation and apologize for the inconvenience. We have now checked the provided web link (<https://www.melvin.pittlabgenomics.com/> & link screenshot below) to ensure it is now functional for multiple individuals and web browsers. It is possible a transient issue on our end rendered this Reviewer unable to access the web page previously. Our sincere apologies if that was the case. Also, as a result of this comment, we have now clarified that the demonstrations provided within this web page

(<https://www.melvin.pittlabgenomics.com/docs/examples>) are additional animated videos. That is, they are not the same as the live demonstration videos provided as Supplementary Movies for this manuscript. We have now clarified this in the **Main Text (Line 186)** and **Supplementary Information (Line 296)**. We thank this Reviewer for helping us avoid this confusion.

<https://www.melvin.pittlabgenomics.com>

2) A git repository was made available by the authors. This should be highlighted in the manuscript, to show that Melvin is not only freely available to use, but can also be potentially extended by other researchers.

We thank the Reviewer for this suggestion. Within our original manuscript, we did provide a GitLab link (https://pittgenomics.gitlab.io/melvin_docs/). The purpose of this link was to provide documentation to users on how to use Melvin. We initially did not provide Git repositories to the Melvin codebase as we were considering possible commercial opportunities. However, after further consideration, we agree with the Reviewer and believe the biggest opportunity is to enable the broader community to re-use or extend the Melvin framework. As such, we have provided Melvin's Alexa Voice Interaction Model (Melvin Alexa Skill) and Skill Service (Melvin Alexa Intent Handler) as public GitHub repositories. These links (provided below) can be found in the **Code Availability** section of the revised manuscript and are referenced in **Line 146** of the **Discussion**.

- https://github.com/pittlab-genomics/Melvin_Alexa_Skill
- https://github.com/pittlab-genomics/Melvin_Alexa_Intent_Handler

3) More space could be devoted in the main text to the specific use cases for a voice assistant? The idea expressed in Sup Fig 5 of use in a tumor board meeting is interesting that could be expanded on. In other words – when, specifically, do the authors see an advantage to a voice assistant rather than a GUI-based approach? Is it that it can be faster for navigation, or are there other benefits?

Thank you for allowing us an opportunity to expand on this. We only provided sparse details within the original manuscript in an effort to adhere to word limitations. In light of recent technological developments around GUI conversational agents (i.e. this Reviewer's Comment #4), we agree that highlighting the value of voice over GUIs is important. Within the original manuscript and this revision (**Lines 40-44**), we mentioned three technical advantages VUIs offer over standard GUIs. One of those advantages – a minimal learning curve – is admittedly now shared by advanced GUIs such as ChatGPT. However, we believe VUIs still offer two key technical advantages for data exploration even with the advent of ChatGPT-like GUIs:

- a) As this Reviewer mentions, faster navigation – regardless of research or clinical settings – is a crucial benefit. This includes minimizing the time to an initial result as well as faster navigation to subsequent results (as demonstrated in **Figure 2**). For both English and

Mandarin, users are able to provide input nearly three times faster via speech-to-text than manual typing¹. We have included the latter point in **Lines 42-43** of this revision.

- b) Due to their strong Internet of Things (IoT) presence, voice-based conversational agents such as Alexa can be quickly accessed from nearly anywhere via mobile phones, smart home devices, and PCs/tablets/TVs that support the Alexa app.

Within the **Discussion**, we now expand upon some domain-specific opportunities for voice-based conversational agents in genomics and precision oncology. As mentioned by this Reviewer, **Supplementary Figure 5** highlights how Melvin could be utilized within the molecular tumor board setting. We agree that our elaboration of this use case in the original manuscript was insufficient. In **Lines 122-132**, we have highlighted the specific benefits Melvin offers within this setting (e.g. determining if certain genomic lesions found in a patient are commonly found in a given cancer type or are actionable). We also elaborate on a future vision for voice-based conversational agents in molecular tumor boards and scientific meetings. Within these settings, these agents could be engaged by any attendee, capture the context of group conversations, and even play a more proactive role (e.g. suggesting and/or displaying information without an explicit prompt). Importantly, potential security concerns related to persistently active listening devices, which would be required for this future vision, are now also discussed (**Lines 133-139**).

4) A bit of an elephant in the room are all the recent advances with Large Language Models are conversational agents (ChatGPT, Bard, etc). Although this work represents a significant contribution without uses of these agents, it may be helpful to consider their impact on future development in the voice assistant space.

We appreciate this suggestion as it is an important consideration for both users and developers. As this Reviewer likely suspects, we have given a great deal of thought to the impact of ChatGPT, Bard, and large language models (LLMs) in general on conversational agents. While ChatGPT and Bard offer an extra layer of sophistication in terms of free text queries and conversational complexity, we currently see these approaches as complementary – with voice-based agents (including Melvin) having the following advantages:

- a) Interface - As mentioned in response to this Reviewer's Comment #3, voice-based conversational agents currently offer faster navigation and broader accessibility relative to these mainstream LLMs.
- b) Information - Perhaps more specific to Melvin than voice-based agents in general, users can obtain vetted, quantitative data that can be subjected to on-the-fly statistical analyses. Below we provided a snapshot of ChatGPT 4.0 where we requested, "Show me [the] percentage of breast cancer patients having TP53 mutations in [the] TCGA dataset. Cite sources as well." While ChatGPT understood the request, it expressed that it does not have access to databases and directed us to the NCI Genomic Data Commons for this information.

Regarding interface, we believe it is inevitable that voice assistants will be connected to more sophisticated ChatGPT-like LLM back-ends to facilitate deeper conversations. Regarding

information, we expect future tools to implement hybrid approaches where LLMs can determine user intents (e.g. the ChatGPT screenshot below) and followed by database queries to perform quantitative analyses. These hybrid approaches will be critical to our proposed molecular tumor board and/or scientific meeting framework (see response to Reviewer 1, Comment #3). We now elaborate on these important points in the **Discussion** from **Lines 127-132**.

✦ GPT-4

 Show me percentage of breast cancer patients having TP53 mutations in TCGA dataset. Cite sources as well.

 As of my last training cut-off in September 2021, I don't have real-time access to specific current percentages in the TCGA (The Cancer Genome Atlas) or any other databases. However, TP53 mutations are commonly found in various cancer types, including breast cancer. The exact percentage of breast cancer patients with TP53 mutations may vary based on the population studied and the specifics of the dataset.

To find the most up-to-date percentage of breast cancer patients with TP53 mutations in the TCGA dataset, you would need to consult the TCGA database directly or refer to recent scientific literature. The National Cancer Institute's Genomic Data Commons (GDC) provides access to TCGA data, which can be accessed and analyzed to obtain this specific information: GDC Data Portal.

However, if you're looking for previously published literature on the topic, you can look up articles in PubMed or Google Scholar that reference TP53 mutations in breast cancer using TCGA data. It's worth noting that the percentage could differ depending on the specific cohort within the TCGA dataset, the specific type of breast cancer analyzed, and the method used to detect mutations.

If you want a precise and current percentage, I would recommend directly accessing the TCGA database or reviewing recent literature on the topic.

 Regenerate

Send a message 
ChatGPT may produce inaccurate information about people, places, or facts. ChatGPT August 3 Version

5) I find the idea of the pronunciation quiz to be very interesting. Perhaps a further avenue of research to be in repurposing this data (for example, so that other voice assistants correctly pronounce some of these terms in the most logical way).

We are happy that this Reviewer also sees this as an opportunity. In early development, we explored adjusting Melvin’s voice responses so that gene name pronunciations better reflect canonical, human pronunciations. However, due to limited manpower, this line of research was downgraded in priority relative to other Melvin features. Nonetheless, in addition to expanding crowdsourced utterance data so that voice assistants can better “hear” users (i.e. Reviewer 1, Comment #6), we agree there is an opportunity to improve Melvin’s spoken responses – as well as those of voice assistants in general. Within the **Discussion at Line 120**, we now mention this as a valuable, potential research effort. We have also provided the code bases supporting the Pronunciation Quiz skill and intent handler as public git repositories, and their web links are provided in the **Code Availability** statement.

6) It may be worth noting the limitations of the crowdsourced utterance data, which perhaps overlooks some major categories of accent due to the particular focus (American regional accents, British accents, etc)

This is a great point, and we agree that there are limitations to our crowdsourced utterance data. In fact, the Melvin Recorder Service was developed as an effort to circumvent these potential limitations. However, we now recognize that this reasoning was unclear in the original manuscript. In the **Results (Line 107)** of the revised manuscript, we now more clearly mention that the purpose of the Melvin Recorder Service was to combat potential limitations of the Out-of-vocabulary Mapper Service (OOVMS). When first describing our crowdsourced utterance data in the **Results (Line 99)**, we now mention that these domain experts were of mixed nationalities and reference the Methods where readers can find more information – including the actual nationalities of the crowdsourcing agents. Lastly, within the **Discussion (Line 119)**, we now mention that richer sampling of national and regional accents could further improve the OOVMS.

7) Does the use of this design of state machine with particular slots make the skill challenging to extend? How would the authors propose extending it to include additional data sources? Please add a few comments about how the skill can be improved in the future.

Thank you for this important question. It is true that a state-based design could be problematic – particularly for analytics domains where data features and/or their values are not well defined. However, given the consistent data structure within the cancer genomics space, we have found it to be quite powerful in terms of extensibility. For example – as mentioned in response to Reviewer 2 below – the Melvin state machine has been implemented with an existing `DATASET` slot, which defaults to The Cancer Genome Atlas (TCGA). We have added an additional breast cancer dataset (BASIS, n = 344) to highlight how Melvin – even in its existing form – can be extended to other cancer genomics datasets (see **Results at Lines 66-69**). As shown in **Supplementary Movie 2**, users can now access this dataset by saying, “Show me BASIS.” Melvin will facilitate all BASIS queries involving mutations (SNVs and indels) and copy number alterations (amplifications & deletion). We have validated the accuracy of the underlying calculations and details on the BASIS dataset (data sources, variant filtering, etc.) have been added to the **Methods (Lines 270-279)**. Importantly, as this dataset does not include gene

expression data, it also demonstrates Melvin's ability to gracefully handle scenarios where a Melvin-supported DATA TYPE is not present in a DATASET. Within the **Discussion**, we now discuss the following points:

- a) Our inclusion of BASIS demonstrates Melvin's ability to ingest and allow exploration of other cancer genomics datasets.
- b) Our ongoing effort to include the International Cancer Genome Consortium's Pancancer Analysis Working Group (ICGC PCAWG) data in Melvin. Similarly, future development will include statistical analyses across datasets (e.g. comparing the same cancer types between TCGA and ICGC PCAWG).
- c) The Melvin framework is extensible and can support more advanced analytics by expanding the number of possible attributes and intents.
- d) One potential limitation of Melvin is its gene-centric design. The state machine, attribute types, and the resulting output may require some modification if other genomic (e.g. chromosomes) or functional elements (e.g. miRNAs) are to be queryable.

We are grateful for this suggestion, and we hope this Reviewer these points on state machine extensibility and limitations satisfactory.

Thank you for the opportunity to review this research.

This was a very insightful review. We truly appreciate the constructive feedback on Melvin's design principles, limitations, and future applications.

Reviewer #2 (Remarks to the Author):

This is an interesting paper that presents Melvin, a voice-based user interface for exploring cancer genomics data. The authors have developed a flexible architecture and conversation model that allows users to query mutation, copy number, and expression data from TCGA through natural language interactions.

Some limitations to consider:

1. The data source is currently restricted to TCGA, so expanding to additional cancer genomics datasets could improve utility. Providing access to raw sequencing data may also be valuable for power users.

We thank the Reviewer for these helpful suggestions. We agree that the utility of Melvin could be increased by allowing users to query cancer genomic datasets beyond TCGA. While not discussed in detail within the original manuscript, the Melvin state machine already supports `DATASET` as an additional attribute type. As such, the Melvin framework does naturally support expansion to other datasets. Given the recent advances in general-purpose conversational agents such as ChatGPT and Bard, we thought it critical to publish Melvin with TCGA as the most comprehensive cancer genomics dataset and prioritize the ingestion of further datasets post-publication (i.e. within the next year). Nonetheless, this Reviewer's comment makes clear we should demonstrate this capability as a proof of principle and be explicit with Melvin's future data ingestion plans. As such, we have taken the following steps:

1. We have added support for an independent set of samples from the Breast Cancer Somatic Genetics Study (BASIS). This set includes mutational (SNV and indel) and copy number alteration (amplification and deletion) data from 344 individuals profiled via WGS and Affymetrix SNP 6.0 array. These data are described further in Supplementary Information.
2. The addition of BASIS and Melvin's ability to expand to different datasets is now mentioned in the **Results (Lines 66-69)** and **Discussion (Lines 149)**. A demonstration of Melvin's interactions over BASIS is now provided as **Supplementary Movie 2**.
3. Within the **Discussion (Line 150)**, we also mention our intent to continue adding high-value datasets to Melvin such as the International Cancer Genome Consortium's Pan-cancer Analysis of Whole Genomes (ICGC PCAWG).

While designing Melvin, we considered developing functionalities that could provide access to raw sequencing data. However, most of the anticipated bam or fastq operations (e.g. alignment, bam slicing, pileups, etc.) were deemed too computationally expensive to be returned in real-time for an effective conversation. We do agree that there exists an opportunity to leverage conversational agents to help with other day-to-day laboratory operations such as raw data processing.

2. The accessibility of voice interfaces depends on ownership of Alexa devices. Integrating with web or mobile could increase reach.

Thank you for this comment, and we apologize for the confusion. While Melvin does require Amazon Alexa as a voice assistant, it does not require users to own Amazon's Echo devices such as those used in our Supplementary Movie demonstrations. Alexa is also available via the Alexa app that can be installed on iOS/Android/Windows mobile, tablets, personal computers, and even smart TVs. Through this app, users are able to access Melvin using these everyday devices, and we anticipate mobile being one of the primary ways users will interact with Melvin. We now provide information on how to obtain the Alexa app within Supplementary Information. We also recognize that the phrase, "any Alexa-enabled device" within the original manuscript is too ambiguous. We have now clarified this in the revised manuscript (**Lines 51-52**) with the following text:

"Here we present Melvin, a VUI to explore and analyze cancer genomics data using any Amazon Alexa-capable device (e.g. mobile phones, tablets, Amazon Echo, etc.)."

3. More advanced statistical and visualizations capabilities could further augment what users can achieve through conversational interactions.

We appreciate that this Reviewer sees the value of advanced analytics via conversational interactions. As mentioned above (Reviewer 2, Comment #1), we will be adding other high-value datasets such as ICGC PCAWG to Melvin over the next year. The existing *compare* functionality (**Supplementary Movie 5**) can then be used to contrast two datasets of interest – further increasing Melvin's statistical and visualization capabilities.

Overall, as highlighted in **Figure 2**, our state machine design underlies Melvin's approach to analytics. Melvin automatically returns analyses and visualizations based on the current attributes and their values (i.e. state). Here, we had to balance analytical detail with concise, conversational responses and fast navigation. Once at a particular state, it is possible to perform deeper, state-specific intents. State-specific intents do not alter state attributes or values. Instead, they request further information about the attributes/values of the current state. Melvin's *gene definition* and *drug options* (**Supplementary Movie 3**) are examples of state-specific intents that can only be invoked after a user navigates to a state containing a gene. We further demonstrated the power of state-specific intents with our *compare* and *split-by* functionalities (**Supplementary Movie 5**). The state machine framework – through additional attribute types, heuristics, and state-specific intents – can be further leveraged to support any number of advanced statistical operations and visualizations. However, since Melvin is a novel interface that already provides comparable functionality to GUI competitors such as cBioPortal, we were apprehensive about developing additional analytical features without clear user demand.

Nonetheless, we recognize that Melvin's natural extensibility should be made more explicit in the main text, and we greatly appreciate this Reviewer's comment for bringing it to our attention.

This is now highlighted in the revised manuscript (see **Discussion** at **Lines 140-148**). We hope this will encourage users to request or – now that Melvin’s interaction model is publicly available – develop more advanced statistical/visualization features.

4. Discussion of privacy considerations is limited. As voice data presents unique risks, this should be expanded.

This is an excellent point, and we appreciate this Reviewer for bringing it to our attention. Within the original manuscript, we mentioned molecular tumor boards as a potential clinical use case for Melvin. In the **Discussion (Lines 133-139)**, we now elaborate on potential security considerations such as private information being overheard by Alexa-capable devices. Additionally, we recognize that there are general concerns around voice assistants like the storage of users’ voice recordings. As such, it is important for us to clarify that Melvin does not carry notable risks beyond those already inherent to Amazon Alexa. Voice utterances that are received by Echo devices or the Alexa app immediately undergo speech-to-text conversion by Amazon-managed servers, which are not accessible to skill developers. Only the resulting transcribed text – and not audio recordings – are received by the Melvin service. As mentioned within Melvin’s Terms of Use (https://pittgenomics.gitlab.io/melvin_docs/docs/policies/terms), it is possible that Amazon will retain audio in accordance with Alexa’s own Terms of Use. This is now mentioned explicitly in the **Discussion (Lines 136-137)**, and we direct readers to the Supplementary Information for further details. These details include a link to Melvin’s Terms of Use (**Supplementary Note 9.2**) and information on how to delete audio recordings from your Amazon Alexa account (**Supplementary Note 1.7**). We thank this Reviewer for the suggestion, and we hope this transparency allows users to make more informed decisions regarding their use of voice technology – including Melvin.

Overall, this is a promising proof-of-concept for conversational AI in genomics. The multi-turn model moves the field forward and the technical innovations around speech recognition are impactful. Addressing some of the limitations around data sources, platforms, analytics, and privacy could strengthen the work further.

We genuinely thank this Reviewer for their supportive comments and helpful insights/suggestions.

References

1. Ruan, S., Wobbrock, J. O., Liou, K., Ng, A. & Landay, J. A. Comparing Speech and Keyboard Text Entry for Short Messages in Two Languages on Touchscreen Phones. *Proc. ACM Interact. Mob. Wearable Ubiquitous Technol.* **1**, 1–23 (2018).

REVIEWERS' COMMENTS:

Reviewer #1 (Remarks to the Author):

Thank you sincerely for your revision of this manuscript. I appreciate the effort that was put in by the authors to address the raised concerns and believe that the resulting manuscript is quite strong. I support the advancement of this manuscript to publication.

Reviewer #2 (Remarks to the Author):

I have no further comments. Thank you.

REVIEWERS' COMMENTS:

Reviewer #1 (Remarks to the Author):

Thank you sincerely for your revision of this manuscript. I appreciate the effort that was put in by the authors to address the raised concerns and believe that the resulting manuscript is quite strong. I support the advancement of this manuscript to publication.

Reviewer #2 (Remarks to the Author):

I have no further comments. Thank you.

We are grateful to the Referees and editorial team for a helpful and smooth review process.